# POLB Regulates Proliferation and Apoptosis of Bovine Primary Myocytes

**DOI:** 10.3390/ani14091323

**Published:** 2024-04-28

**Authors:** Geyang Zhang, Jiamei Wang, Yulong Li, Zijing Zhang, Xiangnan Wang, Fuying Chen, Qiaoting Shi, Yongzhen Huang, Eryao Wang, Shijie Lyu

**Affiliations:** 1Institute of Animal Husbandry and Veterinary Science, Henan Academy of Agricultural Sciences, Zhengzhou 450002, China; 15093389343@163.com (G.Z.); liyulonghaha@outlook.com (Y.L.); vincezhang163@163.com (Z.Z.); 18838917969@163.com (X.W.); fychen2004@sina.com (F.C.); sqtsw@126.com (Q.S.); 2College of Animal Science and Veterinary Medicine, Henan Agricultural University, Zhengzhou 450046, China; 3College of Animal Science and Technology, Northwest A&F University, Yangling 712100, China; wjm20210409@163.com (J.W.); hyzsci@nwafu.edu.cn (Y.H.); 4The Shennong Laboratory, Zhengzhou 450002, China

**Keywords:** *POLB*, myoblasts, RNA-seq, qPCR, analysis

## Abstract

**Simple Summary:**

This study focused on the DNA polymerase β (*POLB*) gene, which belongs to the DNA polymerase X family involved in DNA replication, repair, recombination, and cell cycle regulation. POLB was identified as a muscle development-related gene through gene screening and gene expression analysis methods. Further validation in bovine primary myocytes showed that overexpression of POLB promoted apoptosis, while gene knockdown had no significant effect. Analysis of related genes revealed that POLB overexpression affected the expression of the CASP9 gene, which is involved in the apoptotic pathway.

**Abstract:**

DNA polymerase β (DNA polymerase beta (*POLB*)) belongs to a member of the DNA polymerase X family, mainly involved in various biological metabolic processes, such as eukaryotic DNA replication, DNA damage repair, gene recombination, and cell cycle regulation. In this study, the muscle development-related gene *POLB* was screened by selection signature and RNA-seq analysis and then validated for the proliferation and apoptosis of bovine primary myocytes. It was also found that overexpression of the *POLB* gene had a pro-apoptosis effect, but interfering with the expression of the gene had no significant effect on cells. Then, the analysis of related apoptotic genes revealed that *POLB* overexpression affected CASP9 gene expression.

## 1. Introduction

Jiaxian Red cattle, a Chinese native cattle breed, are deeply praised by Chinese animal husbandry experts for their high meat quality, tough feeding resistance, and high fertility. A high level of genomic diversity and low inbreeding has been observed in Jiaxian Red cattle [1]. Angus cattle is a world-famous beef cattle breed, with the characteristics of high muscularity and a fast growth rate [2]. Since the twentieth century, breeders have started making enormous changes in the growth, stature, and body composition of American Angus cattle through artificial selection. It has become one of the major varieties of high-grade beef production in the United States, Australia, and other countries. The high selection pressure for fast growth can result in an increase in the frequency of beneficial alleles within Angus cattle. This long-term selection leaves behind selection signatures in the genome around the gene that contributed to the selection response [3].

Previously, we used fixation index (F_ST_) and cross-population extended haplotype homozygosity (XP-EHH) methods to identify the candidate signatures of positive selection in Jiaxian Red cattle by comparing them with Angus cattle [1]. The genes within the regions of positive selection in Jiaxian Red cattle were mainly related to meat quality traits and the immune system response [1]. Comparative genomic analysis between Angus cattle and Jiaxian Red cattle makes it possible to identify selection signatures and candidate genes for growth. In this study, the candidate signatures of positive selection in Angus cattle were identified, which we carried out to obtain more regions and genes that were potentially responsible for growth traits.

RNA-seq analysis has been proven to be an effective method to identify genes for economic traits in cattle [4,5,6]. The transcriptome of subcutaneous adipose tissue in Qaidaford cattle, cattle–yak, and Angus cattle were analyzed by RNA-seq. A total of 4167 differentially expressed genes (DEGs) were identified in cattle–yak adipose tissue and 3269 genes in Qaidamford cattle, compared with those in Angus cattle. Some genes are involved in the PI3K–Akt and ECM–receptor interaction pathways, which are important for lipid metabolism-related biological processes [7]. Combining RNA-seq analysis and selection signature analysis might improve the efficiency of identifying the candidate genes for certain traits.

In this study, the *POLB* gene was screened by selection signature and RNA-seq analysis. *DNA polymerase β* (DNA polymerase beta (*POLB*)) is a DNA polymerase isolated from the bovine thymus [8,9]. *POLB* belongs to the DNA polymerase X family, which plays an important role in DNA replication, DNA damage repair, and cell cycle regulation. *POLB* is used to repair DNA damage by participating in the base resection repair (base excision repair (BER)) system. If DNA damage cannot be effectively repaired in cells, it can easily lead to cell senescence, apoptosis, and carcinogenesis [10]. The effects of the *POLB* gene on the proliferation and apoptosis of bovine primary myocytes were then validated in the current study. This study provides a potential reference for further understanding the regulatory mechanisms of muscle growth and development.

## 2. Materials and Methods

### 2.1. Animals and Sequencing

Blood samples were collected from thirty Jiaxian Red cattle from the core breeding farm of the Jiaxian Red Cattle Breeding Center. Genomic DNA was extracted using the standard phenol–chloroform method and then used for whole-genome sequencing (DNA-seq). Additionally, the whole-genome sequencing data of fifteen Angus cattle were used [1,11]. The read-mapping and SNP-calling procedures were the same as those in a previously performed study [1]. In brief, the Burrows–Wheeler aligner BWA-MEM (v0.7.13-r1126) was used to align the clean reads to the cattle reference genome ARS-UCD1.2. Picard tools were used to filter duplicates, and GATK was used for calling the SNPs. After SNP calling, we used the parameters “variant confidence/quality by depth > 2.0, FS < 60.0, RMS mapping quality > 40.0, MQRankSum > −12.5, ReadPosRankSum > −8.0 and SOR < 3.0” and the mean sequencing depth of variants (all individuals) “<1/3× and >3×” to filter the SNPs.

### 2.2. Selection Signature Analysis

F_ST_ and XP-EHH were used to identify the candidate signatures of positive selection in Angus cattle. The procedures for calculating the F_ST_ and XP-EHH values have been previously described [1]. In brief, F_ST_ analysis was calculated in 20 kb steps in a 50 kb window using VCFtools [12]. XP-EHH statistics based on extended haplotypes were calculated using selscan v1.1 [13]. A positive XP-EHH score suggested that selection was likely to have happened in Angus cattle. The overlaps between each method’s top 1% genomic regions were considered candidate signatures of positive selection in Angus cattle. In addition, the overlaps were checked between these identified regions and known growth-related QTLs available in the Animal QTL Database (Animal QTLdb, release 40) [9]. These overlaps were considered as the candidate regions for cattle growth.

### 2.3. RNA-Seq

RNA-seq data of the longissimus dorsi muscle of Angus cattle (GSE57327) were obtained from the Gene Expression Omnibus (GEO) database. In this dataset, the longissimus dorsi muscle of three Angus cattle at 18 months of age were collected for RNA-seq [14].

Five Jiaxian Red cattle at 18 months of age were slaughtered for tissue collection. The longissimus dorsi muscle near the 13th or 14th rib was manually dissected from each animal immediately. These samples were then stored in liquid nitrogen until use. Total RNA was isolated from each sample using the TRIzol reagent (Invitrogen, Carlsbad, CA, USA). The concentration and quality were measured using the NanoDrop 2000 spectrophotometer (Thermo Fisher Scientific, Carlsbad, CA, USA). RNA integrity was assessed using the Agilent 2100 Bioanalyzer (Agilent Technologies, Santa Clara, CA, USA). The libraries were constructed using the NEBNext^®^ UltraTM RNA Library Prep Kit for Illumina^®^ (NEB, Ipswich, MA, USA) following the manufacturer’s instructions and then subjected to 150 bp paired-end sequencing with an Illumina Novaseq 6000 platform (Illumina, San Diego, CA, USA).

After removing the adaptor sequences and low-quality reads, the clean reads were aligned to the bovine genome (ARS-UCD 1.2) using STAR [15]. HTSeq [16] was used to count the reads mapped to each gene. Fragments per kilobase per million mapped fragments (FPKM) were used to determine the gene expression levels. Differential gene expression analysis of two groups was performed using the DESeq2 package in R [17]. *p*-values were adjusted for multiple testing by estimating the false discovery rate (FDR) using the Benjamini–Hochberg method. Genes with a fold change of >2 and FDR of <0.05 were assigned as differentially expressed. The sequencing and analysis were performed by Novogene Bioinformatic Technology (Beijing, China).

GOseq software (release 2.12) was used to analyze the GO enrichment of different expression genes in the Jiaxian Red cattle and Angus cattle breeds [18]. KEGG pathway enrichment analysis of differentially expressed genes was performed by KOBAS 2.0 [19].

### 2.4. Prioritization of the Candidate Genes

Genes within the identified candidate regions were extracted and prioritized based on the differential gene expression analysis with RNA-seq. Genes that were expressed at an FPKM of >1 in either group and with a fold change of >2 and FDR of <0.05 were prioritized. The *POLB* gene was prioritized for further analysis. Expression of the *POLB* gene in twelve tissues was checked in the Animal Omics Database [19].

### 2.5. Cell Culture

Bovine primary myocytes were isolated from the bovine dorsal longest muscle of the fetus. The cells were cultured in high-glucose Dulbecco’s modified Eagle’s medium (DMEM) with 20% fetal bovine serum (Gibco, Grand Island, NY, USA) and 1% penicillin–streptomycin (HyClone, Logan, UT, USA). All the cells were cultured in a humidified 5% CO_2_ incubator at 37 °C.

### 2.6. Vector Construction, shRNAs, and Transfection

In order to overexpress *POLB*, the full length of the CDS region of the *POLB* (NC_037354.1) gene was amplified using the cDNA of the cattle longissimus dorsi muscle. The overexpression vector was then constructed by inserting SmaI and XmaI digestion sites at both ends of the full-length sequence of the amplified *POLB* and attaching them to pHBAd-MCMV-GFP. Short hairpin RNA (shRNA) (F:5′-gatccTCGCAAACTTTGAGAAGAACGTGAATTCAAGAGATTCACGTTCTTCTCAAAGTTTGCGATTTTTTg-3′,R:5′-aattcAAAAAATCGCAAACTTTGAGAAGAACGTGAATCTCTTGAATTCACGTTCTTCTCAAAGTTTGCGAg-3′) to target *POLB* was designed to inhibit *POLB* and was synthesized by General Biol (Chuzhou, China). When the cell growth adhesion reached 70–80%, the cells were transfected using the LipofecterTM transfection reagent with the following constructs. 1. *POLB* overexpression (*POLB* Cdna): the *POLB* gene-overexpressing recombinant adenovirus vector plasmid and the pHBAd-BHGlox backbone plasmid were co-transfected to generate artificial chromosomes overexpressing *POLB*. 2. *POLB* knockdown (*POLB* shRNA): cells were transfected with *POLB* shRNA constructs to silence *POLB* expression. 3. Negative control (NC): cells were transfected with a non-targeting scrambled-sequence plasmid as a negative control.

### 2.7. Cell Proliferation Assay

The expression of overexpressed and interfering apoptosis marker genes in treated bovine primary myocytes was detected at the mRNA level and protein level by RT-PCR and Western blotting. Cell proliferation was examined using a CCK-8 Kit and RT-PCR following the research protocol afforded by the manufacturer.

### 2.8. Cell Apoptosis and Cell Cycle Analysis

The infestation was carried out in groups according to the experimental requirements, and the samples were collected after 48 h. Cell cycle and apoptosis were detected by the Cytometry Assay Kit. The CASP9 gene associated with apoptosis was detected with the following primers (Table 1).

### 2.9. Statistical Analysis

All samples contained three biological replicates and three technical replicates. All data were expressed as means ± SD. An unpaired two-tailed Student’s *t*-test with Bonferroni correction was performed using GraphPad Prism 9.0 for the statistical analysis. The differences were considered significant, highly significant, and very highly significant when *p*-value < 0.05, *p*-value < 0.01, and *p*-value < 0.001, respectively.

## 3. Results

### 3.1. Candidate Gene Selection

F_ST_ and XP-EHH tests were performed to detect signatures of positive selection in Angus cattle. Based on the analysis, 435 regions of positive selection in Angus cattle were obtained [Figure 1a]. Among these regions, 10 regions overlapped with the known growth-related QTLs that are available in the QTLdb (release 40), involving 9 candidate genes located within these regions (Table 2).

By comparing the transcriptomic data of the longissimus dorsi muscle samples from five Jiaxian red cattle and three Angus cattle, 7131 DEGs were screened. Among the 7131 DEGs, 4945 genes were significantly up-regulated and 2186 genes were significantly down-regulated in Jiaxian Red cattle compared with Angus cattle, respectively. The heatmap shows that there were significant differences in the gene expression profiles between Jiaxian Red cattle and Angus cattle. The two varieties were clustered into two independent groups on the heatmap, indicating that they had distinct genetic differentiation at the transcriptome level. It is worth noting that most differentially expressed genes were expressed more in Jiaxian Red cattle than in Angus cattle. [Figure 1b]. The results demonstrate the significant differences in gene expression between Jiaxian Red cattle and Angus cattle. The GO functional enrichment analysis showed that the differentially expressed genes were mainly enriched in the biological processes related to metabolism. The KEGG signaling pathway analysis showed that the differentially expressed genes were mainly enriched in the oxidative phosphorylation pathway [Appendix A.

In the analysis of the expression of genes in the selective sweeps in Angus cattle compared with Jiaxian Red cattle (Table 3), the *POLB* gene was highly expressed in the longissimus dorsi muscle in both breeds (FPKM > 1). The *POLB* gene was significantly more highly expressed in Angus cattle (fold change > 2; FDR < 0.05). Based on the Animal Omics Database, the *POLB* gene was most highly expressed in the skeletal muscle among the twelve tissues [Figure 2]. The results suggest that the *POLB* gene might be involved in the regulation of bovine muscle growth and development. The *POLB* gene was prioritized for further analysis.

### 3.2. Overexpression of POLB Gene Promotes Apoptosis in Bovine Primary Myocytes

Figure 3 shows that overexpression of *POLB* induces apoptosis in primary bovine myoblasts. The mRNA and protein expression levels of the *POLB* gene were significantly higher in the primary bovine myoblasts with *POLB* overexpression compared with the NC (negative control) group. As demonstrated by RT-PCR and Western blot assays, *POLB* overexpression led to a robust elevation in *POLB* transcripts and protein in bovine primary myocytes, validating the establishment of a *POLB* overexpression cell model for further functional studies. Quantitatively, compared with the NC control, the relative levels of *POLB* mRNA and protein in the overexpressed group were increased by 8.46 times and 3.2 times, respectively (Figure 3a,b) [Appendix A. Compared with the NC group, the cell viability of the primary bovine myoblasts in the *POLB* overexpression group was significantly decreased, which suggests *POLB* overexpression decreased cell proliferation rates (Figure 3c). Moreover, the cell apoptosis rate (Figure 3d) and flow cytometry (Figure 3e) revealed increased apoptosis in *POLB*-overexpressing cells. The percentage of cells in the S-phase was also significantly higher with *POLB* overexpression (Figure 3f,g), which suggests that overexpression of the *POLB* gene can affect the S-phase of bovine primary myocytes.

### 3.3. Knockdown of POLB Gene Does Not Inhibit Apoptosis in Bovine Primary Myocytes

Figure 4 shows that the knockdown of *POLB* did not significantly impact the proliferation or apoptosis of bovine primary myocytes. Compared with the negative control group (NC), the mRNA (Figure 4a) and protein expression (Figure 4b) levels of the *POLB* gene in bovine primary muscle cells were significantly reduced after *POLB* knockdown, to 90.6% and 77.3% of the NC levels, respectively [Appendix A. The knockdown efficiency in bovine primary muscle cells was greater than 70%, which was suitable for subsequent experiments. However, according to the cell viability assay results (Figure 4c), at 24 h, the *POLB* knockdown and NC groups had similar cell viability levels, with no significant difference between them. At 48 h, although there was a certain difference in cell viability between the *POLB* knockdown and NC groups, statistical tests showed that the difference did not reach a significant level. After continuous culture for 72 h, the viability curves of the *POLB* knockdown and NC groups were similar again, with the viability of the two groups returning to comparable levels, and the difference was not significant. Compared with the NC group, *POLB* knockdown did not considerably alter cell apoptosis levels (Figure 4d,e).

### 3.4. Overexpression of POLB Gene Affects Expression of Apoptosis-Related Gene CASP9

Figure 5 shows that overexpression of the *POLB* gene affects the expression of apoptosis-related genes. Only CASP9 mRNA expression was increased in the *POLB* gene-overexpressing bovine primary myocytes (Figure 5a), but no change was observed upon *POLB* knockdown (Figure 5b) relative to the NC. Western blot analysis showed that *POLB* overexpression increased the CASP9 protein level by 2.63 times compared with the NC group (Figure 5c) [Appendix A. Therefore, *POLB* overexpression may activate CASP9-mediated apoptosis, while *POLB* knockdown does not substantially impact this pathway.

## 4. Discussion

In this study, we compared the genome differences between Jiaxian Red cattle and Angus cattle through selective scanning analysis and found that the *POLB* gene was in the QTL region affecting body weight gain, suggesting that the *POLB* gene may be related to the growth traits of cattle. By the functional verification test of *POLB* gene overexpression and knockdown, it was found that *POLB* overexpression induced the apoptosis of bovine primary myoblasts, but *POLB* knockdown did not cause significant changes.

Comparative genomic analyses among different cattle breeds have been successfully used to identify signatures of selection and candidate genes for economic traits. Selective sweep analysis is able to detect different genomic regions between cattle breeds that may contain variations affecting important economic traits [20,21]. In this study, selective sweep analysis was performed to identify candidate regions associated with cattle growth by comparing Jiaxian Red cattle and Angus cattle. By this method, the pleomorphic adenoma gene 1 (PLAG1) was found, which was already proven to play an important role in cattle growth. In the F2 population of Holstein and Jersey cattle, the PLAG1 gene was found to be located in the QTL associated with body weight [22]. PLAG1 can promote proliferation and inhibit the apoptosis of bovine primary myoblasts [23]. We also found the *POLB* gene shown in the genome is different between Jiaxian Red cattle and Angus cattle by selective sweep analysis. In the Animal QTLdb, *POLB* is located within a QTL (no. 69373) associated with the body weight gain of cattle. This preliminary evidence suggested that *POLB* might affect cattle growth. Although the RNA-seq data came from public databases with potential batch effects, we introduced feature selection analysis. This approach helped minimize batch effects and identify relevant growth-related genes.

While analyzing public datasets has the possibility of random findings, our in-depth investigation of *POLB* substantiates its relevance. We characterized *POLB*’s expression patterns across bovine tissues. It showed enrichment in muscle, a key growth tissue. Importantly, we proved overexpression of *POLB* had a pro-apoptosis effect on bovine primary myocytes. In mice, Baguma-Nibasheka found that the expression of the *POLB* gene was significantly higher in the muscle with MyoD gene deletion than in normal mice, suggesting that the expression of the *POLB* gene may be related to muscle development [24]. These results indicate selective sweep analysis is feasible for selecting growth trait candidate genes. *POLB* is a good candidate for further cattle growth research. The bioinformatic prioritization combined with empirical validation underscores *POLB*’s involvement in bovine muscular growth and development. Nonetheless, we acknowledge that our study was limited by the use of public RNA-seq data and a focus on in vitro muscle cell models. Future longitudinal gene expression profiling in cattle at different growth stages and in vivo analyses would elucidate *POLB*’s mechanisms.

As an integral component of the DNA repair mechanism, *POLB* collaborates with oncogenes, tumor suppressors, cell cycle regulators, and apoptosis factors to uphold genomic stability and integrity. Increasing evidence has demonstrated a close association between *POLB* expression and the regulation of cell proliferation, growth, and circulation [25,26]. In this study, we found that overexpression of the *POLB* gene can promote the apoptosis of bovine primary myoblasts, but inhibition of the *POLB* gene has little effect on cells. This was consistent with the results of Yuan Liu’s study, which showed that the synergistic effect of *POLB* and FEN1-terminal nuclease could regulate CAG repeat expansion and, thus, affect apoptosis [27]. Nevertheless, *POLB* knockdown induces cell cycle defects and enhances cell proliferation in mouse esophageal squamous cell carcinoma and human oral squamous cell carcinoma [28,29]. No similar results were observed in our study. The different results between the studies may be due to tissue specificity or cell type. Further studies are needed to systematically elucidate the tissue-dependent role of *POLB* in mediating DNA repair, cell cycle progression, proliferation, and apoptosis. Our study provides preliminary evidence for the involvement of *POLB* in bovine muscle growth and development. Future studies are necessary to explore the translational potential of *POLB*-centric regulatory mechanisms in improving important economic traits in cattle.

CASP9, a member of the caspase family of cysteine proteases, is involved in apoptosis and cytokine processing [30]. The CASP9 protein belongs to the BH3-domain protein family and is one of the important regulatory factors affecting apoptosis. In this study, we found that overexpression of the *POLB* gene can promote the apoptosis of bovine primary myoblasts. At the same time, CASP9 significantly increased. CASP9 is a key executor of apoptosis and plays a key role in both endogenous and exogenous apoptotic pathways. Its activation can directly lead to apoptosis [31]. Therefore, *POLB* may induce the apoptosis of bovine primary myocytes by up-regulating the expression and activity of CASP9. Future studies are needed to verify the interaction between *POLB* and CASP9 and clarify their exact mechanisms of action in the apoptotic network of muscle cells. This will be helpful to better understand the molecular mechanism of the *POLB* gene in regulating muscle cells.

## 5. Conclusions

In conclusion, this study showed that the *POLB* gene plays an important role in the proliferation and apoptosis of bovine primary myocytes. Overexpression of *POLB* in bovine primary myocytes can decrease cell proliferation and promote cell proliferation, while *POLB* knockdown exhibits no significant difference. These findings help us better understand the role and regulatory mechanisms of the *POLB* gene in skeletal muscle formation.

## Figures and Tables

**Figure 1 animals-14-01323-f001:**
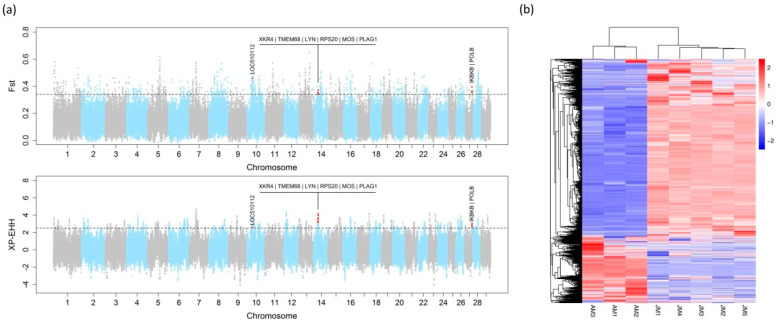
Candidate gene selection based on selection signature and RNA-seq analysis. (**a**) Manhattan plot of selective sweeps in Angus cattle. Genes within the growth-related QTLs are shown. Red dots mean the regions harbored the candidate genes. (**b**) Heatmap showing log2 fold change of the differentially expressed genes between the longissimus dorsi of Jiaxian Red cattle and Angus cattle at 18 months.

**Figure 2 animals-14-01323-f002:**
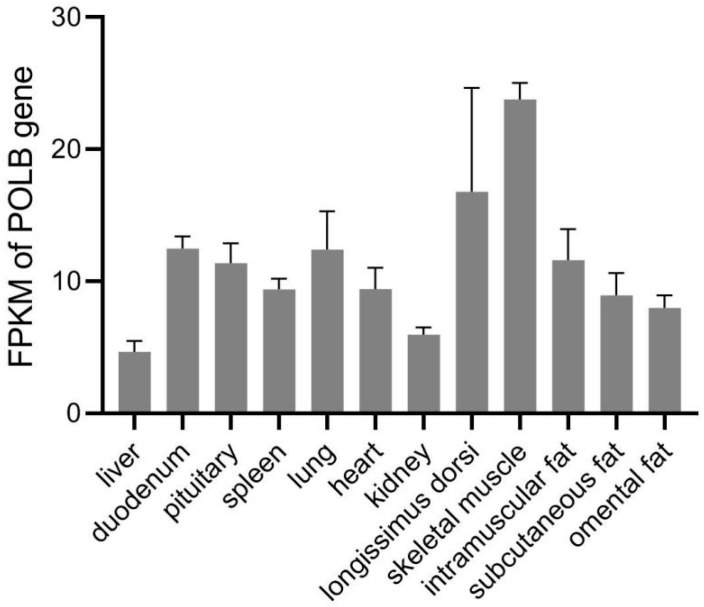
FPKM of *POLB* gene in different tissues of adult cattle in Animal Omics Database.

**Figure 3 animals-14-01323-f003:**
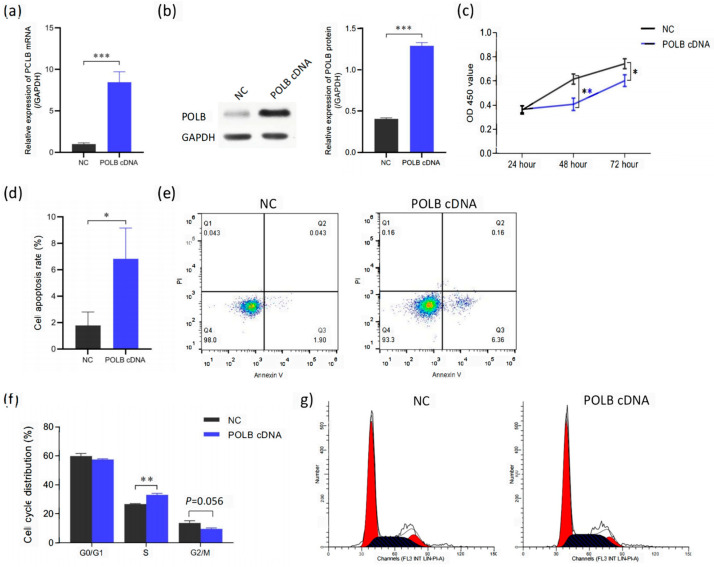
Overexpression of *POLB* gene promotes apoptosis in bovine primary myocytes. (**a**) RT-PCR analysis of the efficiency of overexpression of *POLB* in bovine primary myoblasts. (**b**) Western blot and RT-PCR analysis of the protein expression levels when overexpressing the *POLB* gene. (**c**) Cell viability of bovine primary myoblasts was detected by CCK-8 assay, while *POLB* was overexpressed 24 h later. (**d**,**e**) RT-PCR and flow cytometry analysis of the effects of overexpression of the *POLB* gene on apoptosis. (**f**,**g**) Cell cycle phase index of bovine primary myoblasts was detected by flow cytometry. The data are presented as means ± standard deviation. * *p* < 0.05, ** *p* < 0.01, and *** *p* < 0.001.

**Figure 4 animals-14-01323-f004:**
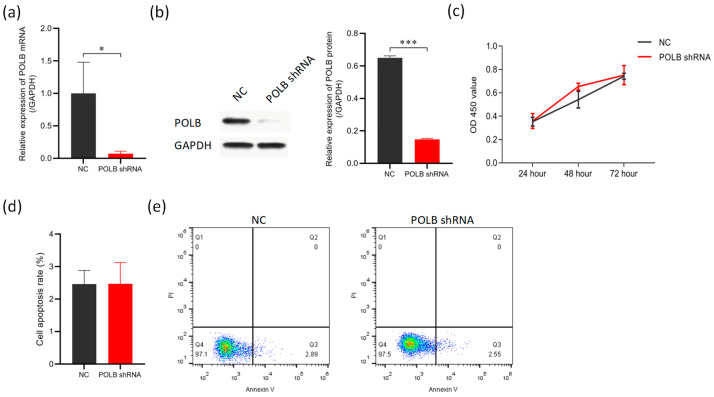
Knockdown of *POLB* gene does not inhibit apoptosis in bovine primary myocytes. (**a**) RT-PCR analysis of the efficiency of inhibition of *POLB* in bovine primary myoblasts. (**b**) Western blot and RT-PCR analysis of the protein expression levels of inhibition of the *POLB* gene. (**c**) Cell viability of bovine primary myoblasts was detected by CCK-8 assay, while *POLB* was inhibited 24 h later. (**d**,**e**) RT-PCR and flow cytometry analysis of the effects of inhibition of the *POLB* gene on apoptosis. The data are presented as means ± standard deviation. * *p* < 0.05 and *** *p* < 0.001.

**Figure 5 animals-14-01323-f005:**
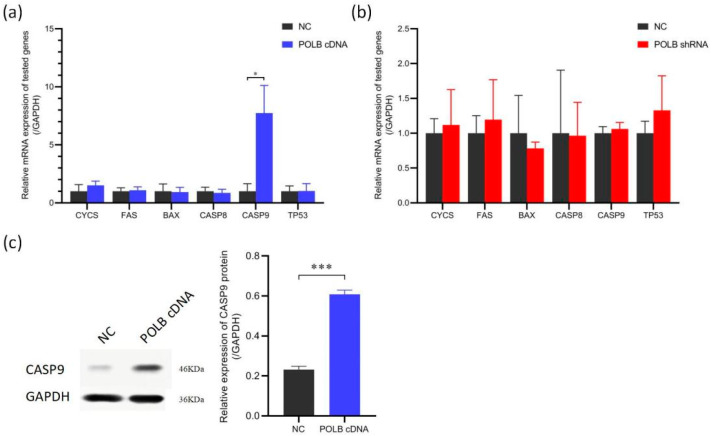
Overexpression of *POLB* gene affects expression of apoptosis-related gene CASP9. (**a**) RT-PCR analysis of the effect of overexpression of *POLB* on other apoptosis-related genes in bovine primary myoblasts. (**b**) RT-PCR analysis of the effect of inhibition of *POLB* on other apoptosis-related genes in bovine primary myoblasts. (**c**) Western blot and RT-PCR analysis of the effects of overexpression of the *POLB* gene on CASP9. The data are presented as means ± standard deviation. * *p* < 0.05 and *** *p* < 0.001.

**Table 1 animals-14-01323-t001:** Primers used in this study.

Gene	Primer Sequence (5′ to 3′)	GeneBank ID	PCR Product (bp)
DNA polymerase beta (POLB)	Forward: AATCCGTCCCCTGGGTGTCA Reverse: GGATGGGCCTCACTCGCTTC	NM_001034764.1	127
Glyceraldehyde-3-phosphate dehydrogenase (GAPDH)	Forward: ATGGAGAAGGCTGGGGCTCA Reverse: GTTGGTGGTGCAGGAGGCAT	NM_001034034.2	153
Cytochrome C, somatic (CYCS)	Forward: TCAGAAGTGTGCCCAGTGCC Reverse: TCAGCGTCTCCTCTCCCCAG	NM_001046061.2	161
Fas cell surface death receptor (FAS)	Forward: GCTCTGCTCAGAGGGGAACG Reverse: GGTGTTGCTCGTTGGTGTGC	NM_174662.2	241
BCL2-associated X, apoptosis regulator (BAX)	Forward: CTCAAGGCCCTGTGCACCAA Reverse: GTCTGCCATGTGGGTGTCCC	NM_173894.1	152
Caspase-8 (CASP8)	Forward: GCCGGCCATGTCAGACTCTC Reverse: TTCAGGCACCTGCTTCCGTG	NM_001045970.2	142
Caspase-9 (CASP9)	Forward: TGGACGCTGGTTCTGGAGGA Reverse: CGCGGCAGAAGTTCACGTTG	NM_001205504.2	135
Tumor protein p53 (TP53)	Forward: CGGAGGTTGTGAGGCGTTGT Reverse: TCCGTCCCAGCAGGTTACCA	NM_174201.2	297

**Table 2 animals-14-01323-t002:** The summary of selective sweeps in Angus cattle compared with Jiaxian Red cattle.

Chr ^1^	Start (bp)	End (bp)	F_ST_ Value	XP-EHH Value	Gene in the Region	Overlap with QTLs Published in QTLdb ^2^
10	27920001	27970000	0.46	2.57	LOC510112 (OR4F13)	Weaning weight (68,115, 68,116)
14	22700001	22750000	0.34	3.29	XKR4	Insulin-like growth factor 1 level (57,469, 57,478)
14	22720001	22770000	0.35	3.16	XKR4	Insulin-like growth factor 1 level (71,532, 30,649)
14	23000001	23050000	0.34	3.66	TMEM68	Insulin-like growth factor 1 level (71,518, 71,519, 71,527)
14	23220001	23270000	0.34	4.15	LYN	Metabolic body weight (131,341, 131,347, 131,348, 131,349)
14	23240001	23290000	0.37	3.96	RPS20; LYN	Metabolic body weight (131,341, 131,347, 131,348, 131,349, 131,351)
14	23280001	23330000	0.34	3.53	MOS; PLAG1	Insulin-like growth factor 1 level (71,513, 71,514), carcass weight (122,423), longissimus muscle area (122,424), and metabolic body weight (131,351)
27	37120001	37170000	0.35	2.68	IKBKB	Body weight gain (69,373)
27	37140001	37190000	0.36	2.92	IKBKB	Body weight gain (69,373)
27	37160001	37210000	0.39	3.00	IKBKB; *POLB*	Body weight gain (69,373)

^1^ Chr represents chromosome; ^2^ Previously identified QTLs related to growth with records in the QTLdb (https://www.animalgenome.org/cgi-bin/QTLdb/BT/index (accessed on 15 February 2020), release 40). IDs of the QTLs are shown in brackets.

**Table 3 animals-14-01323-t003:** Expression of genes in the selective sweeps in Angus cattle compared with Jiaxian Red cattle.

Gene ID	Gene Name	Gene Position	FPKM in Jiaxian Red Cattle	FPKM in Angus Cattle	log2FC	*p*-Value	FDR
ENSBTAG00000003549	OR4F13	10:27939007-27939951	0.00	0.00	/	/	/
ENSBTAG00000044050	XKR4	14:22640221-22953771	0.02	0.00	10.06	0.01	0.03
ENSBTAG00000005893	TMEM68	14:23034280-23070124	5.99	1.05	0.70	0.16	0.29
ENSBTAG00000020034	LYN	14:23134995-23244752	2.84	1.08	0.33	0.57	0.72
ENSBTAG00000019147	RPS20	14:23278316-23279689	177.26	68.45	0.42	0.46	0.62
ENSBTAG00000019145	MOS	14:23299177-23300199	0.00	0.00	/	/	/
ENSBTAG00000004022	PLAG1	14:23330541-23375751	0.19	0.18	−1.00	0.28	0.43
ENSBTAG00000007599	IKBKB	27:37127811-37180362	3.40	1.17	0.57	0.15	0.27
ENSBTAG00000000225	*POLB*	27:37192520-37217387	15.57	2.81	1.50	<0.01	<0.01

## Data Availability

The DNA-seq data of Jiaxian Red cattle are available from GenBank with accession number PRJNA634989. The RNA-seq data of the longissimus dorsi muscle of Jiaxian Red cattle are available from GEO in the NCBI with accession number GSE262289.

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
