# Peer review of "POLB Regulates Proliferation and Apoptosis of Bovine Primary Myocytes"

_animals, 2024, doi:10.3390/ani14091323_

Round 1

Reviewer 1 Report (Previous Reviewer 2)

Comments and Suggestions for Authors

This revised manuscript has been improved.Nevertheless, some concerns need to be solved.

1. Line47 and 49: deg or DEG?

2. LIne 45-51: it is well known that RNA-seq technology has been widely applied for identiying genes related to traits. The author should cited more references to support the point.

3. In the Introduction section, the reason why the POLB gene was chosen as the object of study is not very strong and the logic is not strong?

4. Line 69-74: These sentences should be placed in the section of selection signature analysis.

5. Line 69: Please provide details on the key pipeline and parameters of WGS and SNP calling.

6. The selection signature analysis is key for selecting POLB gene as the object of study. However, I suspect that the authors are not very familiar with the selection signal analysis and are not clear about the key content of the presentation. The description of the methods section supports my speculation. This speculation is also reflected in the identification of SNPs based on WGS.

7. Line 83: why the author chose the version 40? Now, the version 52 is avaiable.

8. For the RNA-seq analysis, why the author didnt consider the batch effect of RNA-seq data?

9. Line 122: please fixed the error of CO2

10. Line 124: please provide the NCBI accession number of POLB genes

11. LIne 150: please provide the threshold level.

12. Line165-169: please rewrite these sentences

13. Line 198: The author should provide the qRT-PCR results to support this point.

14. Figure 3b and 4b, how about the molecular weight of POLB and GAPDH? Also for the Figure 5c

15. Line 229, Why did only CASP9 show a difference between the comparison groups? I doubt the results of the qRT-PCR? Have you attempted to repeat the experiment?

Comments on the Quality of English Language

The language quality of manuscript needs to be improved.

Author Response

Here is our response to the reviewer's comments and the revised manuscript. It should be noted that our transcriptome data has been uploaded to the GEO database, but we have not yet received a login number assigned by the database administrator. Once we have our login number, we will share it with you immediately.

Reviewer 2 Report (Previous Reviewer 1)

Comments and Suggestions for Authors

Report on the manuscript animals-2886102 entitled: POLB regulates proliferation and apoptosis of bovine primary myocytes.

The manuscript has been improved considerably.

Nevertheless, there are still a lot of issues that need be considered before being considered for publication:

-          L. 36-37. Why?
Perhaps, this sentence should be stated after the next paragraph (relationship of Angus and Jianxian Red [1]).

-          L. 147. Table 1 or 2? Where is “Table 1”?
Please, related, review the Table numbering.

-          L. 151-153. More information should be included. Which program was used? Which protocol of statistical analysis? Etc…

-          In the Results section, the description of the results is very scarce. Values defining the observed differences (%?) should be included.

-          Figures 3, 4 and 5:

As previously mentioned by a reviewer, these figures show a large amount of information that has been neither described nor discussed by the authors who only addressed the most basic aspects.

Please, define NC and POLB abbreviations.

Since the results are described as mean ± SD, some of the variables, and for a specific treatment (NC or POLB), showed a huge SD. Such effect needs to be described and considered for discussion.

Some doubts regarding the statistical analysis:

Figure 3c 24 h: please, review the SD bars. More than 2 pairs could be observed in the graphic.
Figure 3d: just p < 0.05?
Figure 3f G2/M: not even a statistical tendency observed?
Figure 4a: just p < 0.05 with such a huge difference?
Figure 4c: For 48 h, was not even a statistical tendency observed?

-          Discussion section has to be rewritten and improved. It lacks scientific validity and accuracy. The discussion only deals with very general aspects. It needs to be improved by including descriptive comments and critical remarks on more specific aspects related to the results obtained and the methodology used.

-          Please, review the journal requirements regarding the “Reference list”. The format of the references does not comply with the journal specifications.

Comments on the Quality of English Language

 L. 25-27. Meaning? Where is the “verb” of that phrase?

-          L. 29. “higher” than… what?

The text is riddled with errors in English. Proofreading is not feasible currently. The text needs to be proofread by a native speaker.

-          Please, review the whole manuscript and fix the italics, and subscripts and superscripts (CO2, etc).

Author Response

Here is our response to the reviewer's comments and the revised manuscript. It should be noted that our transcriptome data has been uploaded to the GEO database, but we have not yet received a login number assigned by the database administrator. Once we have our login number, we will share it with you immediately.

Round 2

Reviewer 1 Report (Previous Reviewer 2)

Comments and Suggestions for Authors

Although the author made changes to the manuscript, the comments I proposed were not effectively addressed. For the selection signature analysis, authors said that the overlaps of each method's top 1% genomic regions were considered candidate signatures. The next sentence said that the genomic regions with a P-value < 0.01 were considered significant targets of selection. So, which threshold level was used in this study? I'm curious as to why the authors used Tajima's D to analyze the genetic diversity of the studied populations. What about the results? I'm not sure if the authors are aware that Tajima's D is also a method for detecting selection signals. Do the authors also use this method to detect selection signals within populations? In summary, I feel that the author's analytical ideas for this study are quite confusing. Meanwhile, I am skeptical about the reliability of the current findings.

Comments on the Quality of English Language

Language editing needs to be improved. 

Author Response

Dear reviewer,

Thank you for your letter and for the reviewers’ comments concerning our manuscript entitled “POLB regulates proliferation and apoptosis of bovine primary myocytes (animals-2886102). Those comments were valuable and helpful for revising and improving our manuscript, as well as for providing important guiding significance to our study. We have studied the comments carefully and made corrections, which we hope will meet with your approval. The revised text is marked in red in the revised manuscript. The main corrections made to the paper and the responses to the reviewers’ comments are as follows:

Reviewer 1

1.Although the author made changes to the manuscript, the comments I proposed were not effectively addressed. For the selection signature analysis, authors said that the overlaps of each method's top 1% genomic regions were considered candidate signatures. The next sentence said that the genomic regions with a P-value < 0.01 were considered significant targets of selection. So, which threshold level was used in this study? I'm curious as to why the authors used Tajima's D to analyze the genetic diversity of the studied populations. What about the results? I'm not sure if the authors are aware that Tajima's D is also a method for detecting selection signals. Do the authors also use this method to detect selection signals within populations? In summary, I feel that the author's analytical ideas for this study are quite confusing. Meanwhile, I am skeptical about the reliability of the current findings.

Response: Thank you very much for your careful review.Sorry for the confusing. Some procedures in the method part were described incorrectly. Only FST and XP-EHH methods were used in this study. Both methods were performed between Angus cattle and Jiaxia Red cattle. The same analysis strategy were used as our previous research. FST analysis was calculated in 50 kb windows with a 20 kb step using VCFtools. XP-EHH was calculated using selscan v1.1. For the results of XP-EHH, a positive score suggests that selection is likely to have happened in Angus cattle, whereas a negative score suggests the selection in Jiaxian Red cattle. The overlaps of each method's top 1% genomic regions were considered candidate signatures of positive selection in Angus cattle. For these two methods, no p-values were generated and then used as a threshold for determining selection signatures. Sorry for another confusion that Tajima's D method was not used in this study either.

We have revised lines 84 to 86 of the “Selection signature analysis” part in the manuscript to make the description more clearly.

Reviewer 2 Report (Previous Reviewer 1)

Comments and Suggestions for Authors

The manuscript has been improved. Nevertheless, again, there are several issues that need to be addressed before being considered for publication.

-          Table 1 and 4 are the same??? 

-          L. 169. Only one short sentence has been used to describe the data shown in Table 3.

Please, improve the description of the results from Table 3 (there are statistically significant p-values, implications?)

-          Fig. 3f G2/M. In my opinion, p = 0.056 should be stated in the figure.

In fact, for all the figures, p-values close to significance should be included in the figures since mean comparison depends on the test used for analysis.

-          L. 187-190 and Fig. 2.

The comment does not agree with the Figure 2 values.

Did the authors carry any statistical analysis? Because “skeletal muscle” might be the highest mean value but “longissimus dorsi” values show a huge SD and therefore, some values would be similar to the highest skeletal muscle…

Please, improve the description of the results L. 187-190.

-          Please, double-check the reference list again. When using an abbreviation for the journal name, a “dot .” must be added after each abbreviation.

Comments on the Quality of English Language

Please, review the whole manuscript. Some typos can be found.

Author Response

Dear reviewer,

Thank you for your letter and for the reviewers’ comments concerning our manuscript entitled “POLB regulates proliferation and apoptosis of bovine primary myocytes (animals-2886102). Those comments were valuable and helpful for revising and improving our manuscript, as well as for providing important guiding significance to our study. We have studied the comments carefully and made corrections, which we hope will meet with your approval. The revised text is marked in red in the revised manuscript. The main corrections made to the paper and the responses to the reviewers’ comments are as follows:

Reviewer 2

1.Table 1 and 4 are the same???

Response: Thank you very much for your careful review.Table 1 and Table 4 are duplicated and we have deleted Table 4.

2.L. 169. Only one short sentence has been used to describe the data shown in Table 3.

Please, improve the description of the results from Table 3 (there are statistically significant p-values, implications?)

Response: Thank you very much for your feedback. After careful check, the original 169 line is indeed the content of table 2, which is the place We marked wrong before. We have corrected "Table 3" on line 169 to "Table 2". For a detailed explanation of Table 3, see the descriptions in lines 185 to 193.

3.-Fig. 3f G2/M. In my opinion, p = 0.056 should be stated in the figure.

In fact, for all the figures, p-values close to significance should be included in the figures since mean comparison depends on the test used for analysis.

Response:Thanks for your pertinent suggestion.We've added "p = 0.056" to Figure 3f G2/M.

4.L. 187-190 and Fig. 2.The comment does not agree with the Figure 2 values.

Did the authors carry any statistical analysis? Because “skeletal muscle” might be the highest mean value but “longissimus dorsi” values show a huge SD and therefore, some values would be similar to the highest skeletal muscle…

Response:Thank you very much for your careful review.The POLB gene exhibited high expression in the longissimus dorsi muscle. Compared to other tissues such as liver, duodenum, pituitary, spleen, lung, heart, kidney, intramuscular fat, subcutaneous fat, and omental fat, the POLB gene showed significantly higher expression levels in skeletal muscle.The detailed data of "FPKM of POLB gene in different tissues of adult cattle in Animal Omics Database" are as follows:

5.Please, improve the description of the results L. 187-190.

Response:We have redescribed this part in lines 184 to 189:" In the analysis of expression of genes in the selective sweeps in Angus cattle comparing with Jiaxian Red cattle (Table 3), POLB gene was highly expressed in longissimus dorsi muscle in both breeds (FPKM >1). And POLB gene was significantly higher expressed in Angus cattle (Fold chang>2, FDR<0.05). Based on the Animal Omics Database, POLB gene was highest expressed in the skeletal muscle among the twelve tissues [Figure 2]."

6.Please, double-check the reference list again. When using an abbreviation for the journal name, a “dot .” must be added after each abbreviation.

Response:Thank you very much for your suggestion.We have added "dot" to the abbreviated journal name.

Round 3

Reviewer 1 Report (Previous Reviewer 2)

Comments and Suggestions for Authors

The molecular weight has not been added to the WB plot yet.

Comments on the Quality of English Language

Not applicable

Reviewer 2 Report (Previous Reviewer 1)

Comments and Suggestions for Authors

The authors have adressed all my comments.

Comments on the Quality of English Language

--

This manuscript is a resubmission of an earlier submission. The following is a list of the peer review reports and author responses from that submission.

Round 1

Reviewer 1 Report

Comments and Suggestions for Authors

Report on the manuscript animals-2694496 entitled: POLB regulates proliferation and apoptosis of bovine muscle cells.

- More information in the Introduction section must be included to justify the originality and relevance of the research in the field.

Animals Journal MDPI aims, and scope are based on animals, including zoology, ethnozoology, animal science, animal ethics and animal welfare.

The authors must address the relationship between their results and the journal scope.

- What does the methodology offer?

Please, describe the pros and cons when comparing to other methodologies.

- L. 254-257. Please, explain the usefulness of the results regarding animal production and/or lean meat yield.

Specific comments:

-          Figure 2 has not been described nor discussed in the manuscript.

-          Figure 2. The huge error bar for LD muscle has to be discussed.

-          Figures 3, 4 and 5. These figures show a large amount of information that has been neither described nor discussed by the authors who only addressed the most basic aspects.

-          Figures 3e and 4e. Statistical analysis? Densitometry?

-          Discussion section has to be rewritten and improved. It lacks scientific validity and accuracy. The discussion only deals with very general aspects. It should be improved by including descriptive comments and critical remarks on more specific aspects related to the results obtained and the methodology used.

Comments on the Quality of English Language

--

Reviewer 2 Report

Comments and Suggestions for Authors

The experimental design of this manuscript is somewhat innovative, and the verification process is step-by-step. However, the manuscript didn't clearly explain the crucial steps of the experiment. For instance, whole-genome sequencing was used to conduct selection signature analysis, but key parameters were missing. The manuscript didn't explain how many SNPs were used for subsequent analysis. RNA-seq is another analysis strategy mentioned in this article, but its results are not mentioned in the results section. In particular, using a single threshold as a condition for identifying differentially expressed genes, which does not meet the requirements for differential gene identification in RNA-seq.

minor comments:

1.Line 38: which breed's chromosome is this?

2. Line 56: DNA-seq?

3.Line 92: How about the threshold levels for the differentailly expressed genes?

4.Line105: please fixed the CO2

5.Line 109: Both SmaI and XmaI should be italic

6.Line111: please provide the sequence of siRNA fragment.

7.Line 127: where is the Figure 1b?

Comments on the Quality of English Language

The experimental design of this manuscript is somewhat innovative, and the verification process is step-by-step. However, the manuscript didn't clearly explain the crucial steps of the experiment. For instance, whole-genome sequencing was used to conduct selection signature analysis, but key parameters were missing. The manuscript didn't explain how many SNPs were used for subsequent analysis. RNA-seq is another analysis strategy mentioned in this article, but its results are not mentioned in the results section. In particular, using a single threshold as a condition for identifying differentially expressed genes, which does not meet the requirements for differential gene identification in RNA-seq.

minor comments:

1.Line 38: which breed's chromosome is this?

2. Line 56: DNA-seq?

3.Line 92: How about the threshold levels for the differentailly expressed genes?

4.Line105: please fixed the CO2

5.Line 109: Both SmaI and XmaI should be italic

6.Line111: please provide the sequence of siRNA fragment.

7.Line 127: where is the Figure 1b?

Reviewer 3 Report

Comments and Suggestions for Authors

The article focuses on the relevant research on the regulation of proliferation and apoptosis of bovine muscle cells by the POLB gene, which is novel and interesting overall. However, there are several points that need minor revision, and it is recommended to accept  after modification.

1. The authors state that the polb gene is located within a QTL related to bovine growth. Has any association analysis been performed in the beef cattle population used in this study?

2. Please specify the meaning of "*" in the figure legends for Figures 3a, 3c, 3f, 4a, 4c, and 5c.

 3. The statement "This suggests that POLB is associated with muscle development in cattle" is too absolute, as this association has not been experimentally verified at this time. Please modify the wording to accurately reflect the preliminary nature of this finding.

 4. The main research findings should be summarized in the abstract and results sections. There is no need to repeat the experimental results in the introduction.

 5. Whether the authors used qRT-PCR to verify the expression levels of POLB in different bovine tissues further verified the effect of POLB gene on bovine muscle growth and development?